# Geriatric Population Triage: The Risk of Real-Life Over- and Under-Triage in an Overcrowded ED: 4- and 5-Level Triage Systems Compared: The CREONTE (Crowding and R E Organization National TriagE) Study

**DOI:** 10.3390/jpm14020195

**Published:** 2024-02-09

**Authors:** Gabriele Savioli, Iride Francesca Ceresa, Maria Antonietta Bressan, Gaia Bavestrello Piccini, Viola Novelli, Sara Cutti, Giovanni Ricevuti, Ciro Esposito, Yaroslava Longhitano, Andrea Piccioni, Zoubir Boudi, Alessandro Venturi, Damiano Fuschi, Antonio Voza, Roberto Leo, Abdelouahab Bellou, Enrico Oddone

**Affiliations:** 1Department of Emergency Medicine and Surgery, IRCCS Fondanzione Policlinico San Matteo, 27100 Pavia, Italy; mita.bressan@gmail.com; 2Emergency Department, IRCCS Humanitas Research Hospital, Via Manzoni 56, Rozzano, 20089 Milan, Italy; irideceresa@gmail.com (I.F.C.); antonio.voza@humanitas.it (A.V.); 3Emergency Medicine, Université Libre de Bruxelles (ULB), 1050 Brussels, Belgium; gaia.bavestrellopic01@universitadipavia.it; 4Medical Direction, IRCCS Fondanzione Policlinico San Matteo, 27100 Pavia, Italy; v.novelli@smatteo.pv.it (V.N.); s.cutti@smatteo.pv.it (S.C.); 5Department of Drug Science, University of Pavia, 27100 Pavia, Italy; giovanni.ricevuti@unipv.it; 6Nephrology and Dialysis Unit, ICS Maugeri, University of Pavia, 27100 Pavia, Italy; ciro.esposito@unipv.it; 7Residency Program in Emergency Medicine, Department of Emergency Medicine, IRCCS Humanitas Research Hospital, Rozzano, 20089 Milan, Italy; lon.yaro@gmail.com; 8Emergency Department, Fondazione Policlinico Universitario A. Gemelli, IRCCS, 00168 Rome, Italy; andrea.piccioni@policlinicogemelli.it; 9Department of Emergency Medicine, Dr Sulaiman Alhabib Hospital, Dubai 2542, United Arab Emirates; zoubir.boudi@gmail.com; 10Department of Political and Social Sciences, University of Pavia, 27100 Pavia, Italy; alessandro.venturi@unipv.it; 11Bureau of the Presidency, IRCCS Fondanzione Policlinico San Matteo, 27100 Pavia, Italy; 12Department of Italian and Supranational Public Law, Faculty of Law, University of Milan, 20133 Milan, Italy; damiano.fuschi@unimi.it; 13Department of Systems Medicine, University of Rome “Tor Vergata”, 00133 Rome, Italy; rtoleo@tiscali.it; 14Global Network on Emergency Medicine, Brookline, MA 02446, USA; 15Department of Emergency Medicine, Wayne State University School of Medicine, Detroit, MI 48201, USA; 16Institute of Sciences in Emergency Medicine, Department of Emergency Medicine, Guangdong Provincial People’s Hospital (Guangdong Academy of Medical Sciences), Southern Medical University, Guangzhou 510080, China; 17Department of Public Health, Experimental and Forensic Medicine, University of Pavia, 27100 Pavia, Italy; enrico.oddone@unipv.it; 18Occupational Medicine Unit (UOOML), ICS Maugeri IRCCS, 27100 Pavia, Italy

**Keywords:** triage, emergency service, hospital, crowding, triage (under-triage), triage (over-triage), geriatric emergencies, overcrowding and access block, overcrowding detection, overcrowding effect, overcrowding, ED management

## Abstract

Elderly patients, when they present to the emergency department (ED) or are admitted to the hospital, are at higher risk of adverse outcomes such as higher mortality and longer hospital stays. This is mainly due to their age and their increased fragility. In order to minimize this already increased risk, adequate triage is of foremost importance for fragile geriatric (>75 years old) patients who present to the ED. The admissions of elderly patients from 1 January 2014 to 31 December 2020 were examined, taking into consideration the presence of two different triage systems, a 4-level (4LT) and a 5-level (5LT) triage system. This study analyzes the difference in wait times and under- (UT) and over-triage (OT) in geriatric and general populations with two different triage models. Another outcome of this study was the analysis of the impact of crowding and its variables on the triage system during the COVID-19 pandemic. A total of 423,257 ED presentations were included. An increase in admissions of geriatric, more fragile, and seriously ill individuals was observed, and a progressive increase in crowding was simultaneously detected. Geriatric patients, when presenting to the emergency department, are subject to the problems of UT and OT in both a 4LT system and a 5LT system. Several indicators and variables of crowding increased, with a net increase in throughput and output factors, notably the length of stay (LOS), exit block, boarding, and processing times. This in turn led to an increase in wait times and an increase in UT in the geriatric population. It has indeed been shown that an increase in crowding results in an increased risk of UT, and this is especially true for 4LT compared to 5LT systems. When observing the pandemic period, an increase in admissions of older and more serious patients was observed. However, in the pandemic period, a general reduction in waiting times was observed, as well as an increase in crowding indices and intrahospital mortality. This study demonstrates how introducing a 5LT system enables better flow and patient care in an ED. Avoiding UT of geriatric patients, however, remains a challenge in EDs.

## 1. Introduction

Triage for geriatric patients is an open challenge for emergency physicians, especially in increasingly overcrowded EDs [1,2,3,4]. In this Special Issue, we focus on the effects of re-thinking the triage system in the geriatric population (>75 years) pertaining to our ED.

Due to an aging population, visits to emergency departments (EDs) by geriatric patients are increasing worldwide. With increasing age, frailty is defined by multiple aspects: among geriatric ED patients, a high proportion have balance difficulties, an unsteady gait, require assistance for mobility, or have decreased muscle strength in their lower limbs. Frail and elderly patients are at risk of adverse outcomes (for example: mortality and prolonged hospital stays). In this respect, suitable triage for fragile geriatric ED patients is a critical healthcare issue.

Waiting times have often been described as improving with the transition to the 5-level triage system [3,4]; however, there is not complete agreement on this in the literature, as on the other hand, a lengthening of waiting times has been described [5]. The various triage systems need to identify and prioritize patients who require an urgent intervention in a short time; therefore, it is important to find solutions with efficacy to reduce waiting times [3,4,5,6,7,8]. For this reason, we went to see how the transition to the 5-level triage system affected waiting times.

In EDs organized by areas of intensity of care, patients at triage receive a priority code and are channeled toward low- or medium-high-intensity care areas [9,10].

It is the common opinion of the authors that 5LTs, due to their increased accuracy and safety, are better than 3LT or 4LT systems. The fact that this advantage also affects geriatric patients has not yet been widely studied [3,8,11,12,13,14,15,16,17,18,19,20,21,22,23,24,25,26,27,28,29,30].

The triage process is complex, and its complexity is even greater for geriatric patients who are more prone to UT [31,32,33,34,35,36,37,38].

Hence, there is a need to study how UT and OT vary in reality during the transition from a 4LT to a 5LT, especially in geriatric people.

Overcrowding compromises the quality of patient care, not just the quality received. However, the reciprocal effects between crowding and triage are still not widely studied. In particular, the influence of the various factors that determine crowding with triage waiting times and the frequency of UT and OT requires investigation. In addition, the outcomes associated with adopting a 5LT system have been explored in the field only by a few studies, and with low numbers of patients [39,40,41,42,43,44,45,46,47,48,49,50,51,52,53].

Some real-life studies have focused on only one symptom or have enrolled a low number of patients. Hence, there is a need to study triage with the complexity of real life in a crowded ED with a large population.

For all these reasons, we analyzed the triage efficacy for geriatric people in a real-life ED, both in a four-level triage system (4LT) and in a five-level triage system (5LT). We studied the waiting times (primary objective) and (secondary objectives) over- and under-triage (OT and UT, respectively). Finally, we analyzed the relationships, in real life, between crowding and triage and the functioning of the 5LT, also during the COVID-19 pandemic.

## 2. Methods

### 2.1. Study Design

We conducted a retrospective study, which encompassed the admittances to the ED of the Foundation IRCCS Policlinic San Matteo from 1 January 2014 to 31 December 2020. During this period, our ED underwent reorganization with a subdivision into areas of different intensities of care and a shift from a 4LT (from 1 January 2014, to 30 November 2015) to a 5LT (from 30 November 2015, to 31 December 2020). The admissions during the two periods were compared.

A tailored investigation was performed to investigate the data of interest. Anonymization was performed in order to ensure confidentiality. The mandatory consent to data utilization for medical and research purposes as well as health data processing, was obtained at admission from all patients.

### 2.2. Endpoints

This investigation was conducted considering the number of patients presenting to the ED during a period of time which was then further subdivided into two periods: the 4LT and 5LT periods. The objective was to ascertain the effects that the introduction of a 5LT system would have on wait times in geriatric (>75 years old) and young populations. The secondary objective was to determine whether adopting a 5LT system had any effect on the accuracy level of the triage of geriatric patients. The accuracy of triage in geriatric people has been measured as the percentages of patients undergoing UT and OT, as well as by verifying the correlation between the code attributed at triage by the triage nurse and the severity code attributed at discharge by a physician.

A further outcome is the correlation between triage and crowding indices. The most robust crowding indices in the literature were used, such as the length of ED stay, total access block time, and rate of access block [54,55,56,57,58,59,60,61,62,63,64,65,66,67,68,69,70,71,72,73,74,75,76,77,78,79,80,81,82,83]. For a detailed definition of the calculation of the same, we referred to our previous publications [84,85,86]. Finally, we analyzed the proficiency of a 5LT system during the exceptional circumstances of the COVID-19 pandemic.

### 2.3. Statistics

Continuous variables are expressed as means, medians, and interquartile ranges; qualitative variables are expressed as the number of observations and appropriate proportions. The non-parametric Mann–Whitney test was used to make between-group comparisons for continuous variables, according to their non-normal distributions. The χ^2^ test was used to study associations between the qualitative variables. The statistical analysis was performed with pertinent logistic multivariate regression models to assess the correlation between time variables while accounting for crowding, exit block, and the different triage periods. The differences in UT and OT by year of observation were examined using the test of proportions. For each passage, the presence/absence of over-triage and under-triage was modeled as a binary variable, as described in the Methods section, and the risk of undergoing either UT or OT was defined as the odds ratio (OR) resulting from multiple regression analysis adjusted for age, gender, and year of observation. The investigation was performed for all the patients presenting in the selected period, as well as for subgroups in which boarding or exit blocks were present. The significance level was set at alpha 0.05 (statistical significance at *p* < 0.05), and all tests were two-tailed.

The analyses were conducted using STATA software (version 14; Stata Corporation, College Station, TX, USA, 2015). The ethics committee submitted and approved the study (protocol number: 20200114609). The analyses were made using data from the PIESSE software (GBIM, Pavia, 2020).

## 3. Results

### 3.1. Overall 

Geriatric patients (>75 years) constituted about 24% of the total population taken into consideration, and their number grew over the years (from 21% to 24%) (Table 1). As can be seen in the table below, the patients became increasingly complex, as can be seen from the triage priority code, priority code at discharge, the need for high care intensity, and outcomes (Table 1).

### 3.2. Wait Time for Geriatric Compared to Younger Patients 

The differences between the wait times of geriatric (>75 years of age) and younger patients had little statistical significance when considering Code 4 and Code 5 (about 1 or 2 min for codes 1, 2, 3, and 5; about 20 min for code 4) (Table 2).

In contrast, with the introduction of 5LT, a constant (even though not statistically significant) increase in wait times of ~3–4 min per year was observed for the patients who were assigned Code 2 at triage (Table 2). This was likely dependent on the increased number of patients receiving a priority Code 2 at triage and crowding at our hospital.

Patients who were assigned Code 3 in the 5LT system period had similar wait times to patients who were assigned a Code 2 in the 4LT system period, with equal tendency for both geriatric patients (26.9 vs. 24.7 min) and younger patients (23.4 vs. 22.5 min) (Table 2). Patients who were assigned Code 4 and 5 in the 5LT system period had wait times which were comparable to those of patients who were assigned Codes 3 and 4 during the 4LT system period (Table 2).

### 3.3. UT and OT in the Geriatric Population 

The risk of UT is slightly greater in geriatric patients than in the general population (OR = 2.22; *p* < 0.001). (Table 3). However, when separately analyzing the areas of low intensity of care (OR = 0.85; *p* < 0.001) and those of medium-high intensity of care (OR = 0.56; *p* < 0.001), this trend was found to be reverted (Table 3 and Table 4). The top three complaints of geriatric patients at triage were minor signs and symptoms (27.3%), abdominal pain (13.1%), and dyspnea (10.5%). These symptoms alone accounted for >50% of admissions. Minor trauma (7.3%) and neurological disorders (6.5%) were less frequently observed (Table 3).

The elderly patients underwent over-triage more often than the general population, and this occurred in both the 4LT and the 5LT system periods, with no difference according to areas of intensity of care (Table 3 and Table 5).

### 3.4. Crowding

The phenomenon of crowding was aggravated over the years following the progressive increase in boarding and exit blocks observed from 2014 to 2020 (Table 6). The total number of ED admissions rose gradually up to 2018, with the exception of a period of decrease in 2015, and then lowered again in 2019 and 2020 (Table 6). The wait times for patients in low and medium-high-intensity care areas (Table 7; *p* < 0.001) increased, as shown by the indices of the boarding and exit blocks. These indices of crowding were chosen due to their higher reproducibility with automated data extraction [84,85,86]. In the 4LT system, boarding substantially correlates with a slight reduction in the rate of under-triage in the low-intensity care area, but with a reversed tendency of increased risk of UT in geriatric patients in medium-high-intensity care areas. When considering the period with a 5LT system, on the other hand, boarding was no longer correlated with a greater risk of UT in elderly patients, and similarly, the probability of UT was also reduced for young people in medium-high-intensity care areas. There is, therefore, a correlation between the phenomenon of boarding, a greater risk of OT in young people, and a lower risk of OT in the elderly. In the 5LT system, this risk of OT in the younger patients was, however, found to be lower, and the risk of OT for elderly patients was also greatly reduced, as shown in Table 8. The exit block correlated with an increase in the rate of under-triage in both 4LT and 5LT. However, switching to 5LT in the medium-high-intensity care area did not increase the risk of UT for young people. During the exit block, there was a clear reduction in the number of over-triage cases, which was an even more marked reduction in the period of 5LT, as shown in Table 9.

When we examined individual triage priority codes for the intensity of the medical examinations and areas of care, we found no statistically significant differences between geriatric (>75 years of age) and younger patients.

### 3.5. LT of COVID Patients

In the period characterized by the COVID pandemic, 3826 patients presented to our ED. 125 patients were assigned a triage Code 5, 2789 a Code 4, 169 a Code 3, 810 a Code 2, and 86 were assigned a triage Code 1. A total of 159 patients tested positive for COVID-19, and 78 infected patients died in the ED. The wait times for patients in the area dedicated to COVID patients were 48 min for Code 5, 47 min for Code 4, 48 min for Code 3, 27 min for Code 2, and 10 min for Code 1.

## 4. Discussion

### 4.1. Overall

The age of patients admitted to the ED has constantly risen through the years. The number of patients presenting to the ED has progressively increased, and patients are currently older, more fragile, and sicker. This trend is in turn reflected by a decrease in spontaneous accesses, a higher number of admissions through the territorial emergency service or on a gurney, and the percentage of more severe priority codes assigned at triage and higher severity codes assigned at discharge. This trend has already been extensively described in the literature [54,55,56,57,58,59,60,61,62,63,64,65,66,74,75,83,84,87,88] and has been negatively influenced by the COVID-19 pandemic, with a greater number of patients requiring hospitalization [85,86]. As a consequence, exit blocks and boarding have increased as well, and together with the progressive reduction in beds, this has led to a worsening of crowding.

All these changes in the population and the flow in the ED have led to a change in the work of ED physicians, transforming their practice from “admit-to-care” to “care-to-admit” [85,86]. This change has resulted in a gradual extension of LOS and processing times. It has already been demonstrated that the increase in crowding has negative effects on patient outcomes and satisfaction; however, until now, the effects of overcrowding on triage—particularly on wait times and on the risk of UT and OT—have not been thoroughly analyzed. The 5LT system has proven itself more accurate, with better correspondence between the code assigned at triage and the severity code assigned at discharge. This therefore suggests that 5LT system triage codes reflect, in a more precise way, the actual acuity of the patient in comparison to 4LT systems. This evidence, together with the data that show a global decrease in the risk of UT, demonstrates that a system that allows a lower risk of UT has clear benefits for the patients in the ED, and especially for those patients who are sicker and more fragile, with a subsequent improvement in the outcome.

### 4.2. Wait Time

Age does not seem to play a significant role in waiting issues. When we corrected age data according to triage code and area of care intensity, our results were in line with what was expected. Some studies correctly described increases in ED crowding due to increases in geriatric access to EDs. As is similar to our findings, Kawano et al. noted that the geriatric population tended to consist of more complex patients who presented more frequently in ambulances. The geriatric frequently required medium-high care intensity and higher triage codes. Therefore, our study is in line with the literature and underscores the finding that age itself does not cause crowding; however, age is a factor related to frailty in compromised patients [89,90,91,92]. The general fragility of geriatric patients increases crowding and wait times. Once age is corrected according to the need for more intense care and faster medical examination, it no longer appears to affect crowding. However, these findings require verification within the context of multicenter studies.

Additional focus should be put on the effects of having both doctors and nurses working at triage. The presence of dedicated doctors at triage allows for a quicker examination of patients coming for less severe problems. This also allows for a prompter evaluation of more-complex patients, helping to reduce the impact that crowding has on ED functioning. The presence of a senior doctor, meaning a medical doctor having obtained the title of specialist in emergency medicine, alongside the triage nurse at the triage station, has positive effects, with reductions in wait time, LOS, LWBS rates, and the percentage of patients leaving without having had a complete workup and treatment. Additionally, triage teams consisting of a doctor and a nurse can also be beneficial for a more rapid admission of more fragile patients (such as geriatric patients) who are not required to stay in the emergency department for further investigations or time-sensitive interventions, but would benefit more from prompt hospitalization. This type of triage team can therefore exert positive effects by reducing overcrowding [93,94,95].

Exit blocks and boarding often influence the wait times of patients who require hospitalization or a secondary transfer, and this phenomenon is prevalent in ED areas with higher care intensity [9,10]. The medium-high-intensity patient requires a set of tools for their care (such as oxygen, non-invasive ventilation, telemetry, and monitoring of vital parameters), and these are dependent on structural limits (i.e., the number of oxygen outlets, ventilators, and monitors). The fact that more and more patients need to be placed in areas of medium-high intensity of care, and the concurrent increase in the phenomenon of exit blocks, result in a saturation of the resources which are available, with a consequent increase in the time needed for patient management.

Increased LOS could also be due to other changes in the internal departmental organization (such as doctor and nurse turn-over, differences in the organization of shifts, etc.). The relationship that exists between boarding times in the ED and patients’ outcomes is still being analyzed. There is a need for larger studies to better analyze the influence that boarding and exit blocks have on adverse ED outcomes [96].

### 4.3. UT and OT in the Geriatric Population

The geriatric population was found to be at increased risk of UT in this study. This tendency remained constant during the 4LT and 5LT system periods, regardless of the area of care intensity to which the patients were allocated [37,97,98,99,100]. This phenomenon may be due to physiological changes during senescence, pain habits, and the inability to communicate [101,102,103,104,105,106,107,108,109,110,111,112,113,114,115,116,117,118,119,120].

The geriatric population is expected to represent ~20% of the overall population by 2030. Importantly, geriatric patients are more complex, require more resources, and have higher admission rates. Geriatric triage is more complex for various reasons. The interpretation of vital signs in the geriatric is more challenging due to homeostatic mechanisms. For example, when considering the respiratory system of an aging individual, the lungs will have less elastic recoil, and there will be increased dead space and a decreased physiologic reserve. Therefore, a respiratory rate >27 breaths/minute in the elderly is an accurate predictor of adverse events and can help to identify the critical patients. When considering the cardiovascular system in the elderly, it is important to note that the combination of myocardial thickening, arterial wall stiffness, and hypertension will result in an increase in the workload of the heart. Elderly patients are more prone to events of orthostatic hypotension due to larger pulse pressures and a reduced effect of circulating catecholamines. A systolic blood pressure <110 mmHg often represents hypotension in geriatric patients, especially among those with traumatic injuries. Resting heart rate also increases with age [101,102,103,104]. Analogous homeostatic changes also occur with the body’s temperature. Elderly patients are also less likely to present with fever because of several factors, such as the presence of a weaker immune system, a decreased cardiac output, and diminished muscle mass. Consequently, slight temperature changes, as well as hypothermia, may represent severe infection in this category of patients [105,121].

In addition to homeostatic changes, several factors that complicate pain assessment should be considered. Elderly people may indeed have altered pain perception; an increased risk of persistent pain; and, when cognitive impairment is present, they might have difficulty in assessing pain and its location [106,107,122,123,124].

Other domains also require special consideration, such as the atypical presentations of common diseases. For example, in geriatric patients, acute coronary syndromes are more likely to present without chest pain. Meanwhile, patients with sepsis may have unaltered parameters and symptoms which are not specific for the identification of the source of the infection. Elderly people with pneumonia often present to the ED without respiratory symptoms, and might not have any chest pain or fever. Some geriatric patients with acute surgical abdomens report only mild pain [108,109].

These data agree with our results. We found that dyspnea and abdominal pain were common in cases of UT. The atypical presentation and communication difficulties can be responsible for a high UT symptomology classified as “minor signs and symptoms”.

Cognitive impairment in this population is also important to consider. Prospective ED studies of patients older than 65 and 70 have evidenced delirium rates of 9.6% and 10%, respectively. Sixteen percent of patients older than 70 years demonstrated an impairment of their mental status, and six percent were found to meet the criteria for both delirium and dementia. A great percentage of patients in the geriatric populations who were affected by delirium were not correctly diagnosed, and several geriatric patients were discharged. Early detection of acute changes in the cognitive behavior at triage and timely transmission of this information to the remaining care team are extremely important [10,85,110,125]. These factors were likely at play among patients who experienced UT in our study.

Polypharmacy in the aged population should also be considered: 44% of US males and 57% of US females older than 65 take 5 or more medications per week. These patients are particularly susceptible to adverse drug events (ADEs). Notably, ADEs account for up to 10% of geriatric ED presentations. Cardiovascular, diuretic, antibiotic, hypoglycemic, sedative, opioid analgesic, anticholinergic, and anti-inflammatory medications are commonly implicated in ADEs [126].

Finally, there is a constant increase in ED visits attributable to falls and trauma among the elderly, with significant morbidity and mortality. However, unlike the younger cohorts, falls are the major trauma mechanism and often occur as a consequence of decreased autonomy, increased fragility, modifications in vision acuity, impaired muscle strength with altered gait and balance issues, an acute medical event, or the introduction of a new medication. In these patients, UT is potentially more frequent due to underestimation of the gravity of the injury as well as of the impact of the comorbidities on the clinical picture [111,126,127]. We found that patients with trauma and minor dynamics often experienced UT. It is important to reduce UT in this category of fragile patients, thereby improving recognition of critical situations. Further studies are needed to investigate possible ways to counteract the increase in OT while keeping the risk of UT to a minimum. Correct triage of geriatric patients, even when adopting 5LT systems, remains an extremely complicated task for triage teams. The implementation of triage algorithms through artificial intelligence could help in overcoming age-related specificities.

### 4.4. Crowding Indices

The increase in LOS, exit blocks, boarding, and processing times provoked a net increase in throughput and output factors, which, in consequence, caused an increase in wait times, both in geriatric people and in young people. This trend was especially observed with exit blocks, more than boarding, in low-intensity areas and for less-urgent triage codes. In these conditions, the presence of an exit block resulted in a wait time which was almost doubled. Additionally, the lengths of wait times for patients who required prompt medical examination and who were assigned high priority codes (triage Codes 1 or 2) increased by approximately 25–30%.

These results confirm the effect of output factors on the flow in the ED. Processing and LOS times are increased due to an increase in the phenomenon of exit blocks, and this in turns influences all ED processes and flows, with a lengthening of wait and handling times.

Exit block allows for a reduction in the risk of OT for those patients assigned to low-intensity care areas and who are given lower priority triage codes, and boarding reduces OT in medium-high intensity care areas. Regarding, O.T.; in both cases, triage is more accurate for patients, for both young and geriatric ones. This may be a consequence of increased vigilance of the triage nurses during crowding. Nevertheless, this increased specificity may be dependent on the fact that during prolonged wait times, patients can undergo re-evaluations more often. The OT reduction is greater in areas of medium-high intensity of care because it is likely that, in these patients, the attention of triage nurses is concentrated in cases of crowding.

During crowding represented by boarding and, even more, by exit blocks, the under-triage worsens: the situation of overcrowding caused by the output factors therefore causes a reduced accuracy of the triage with regard to the under-triage. This reduction in accuracy is probably also due to the lengthening of waiting times in the context of increased crowding. It is interesting to note that 5LT proved to be more effective at minimizing UT risk in the medium-high intensity area where, for young people, there is even a reduction in the risk of UT. It is, therefore, the opinion of the authors that the increased risk of UT is not only due to a lengthening of waiting times with a consequent increase in the stress of triage nurses, but could also be due to an attitude of greater attention paid to the most acute patients in crowding conditions, those destined for areas of medium-high intensity of care. This attitude is more pronounced in younger patients and much reduced in elderly patients, perhaps because of the greater insidiousness of the acute manifestation of geriatric patients. The reasons for a more insidious manifestation in geriatric patients were reviewed in the previous paragraph (change in homeostatic processes, lower reliability of some vital parameters, difficulty in expressing symptoms by the geriatric patient, presence of dementia, etc.).

The study, therefore, shows a greater susceptibility of geriatric patients during crowding. The use of artificial intelligence algorithms could reduce this risk, thus outlining a need for studies in this sense for the future.

Finally, it should be noted that the doubling of crowding represented in Table 4 was simultaneous to the beginning of the COVID-19 epidemic (2020), therefore putting a further strain on the system. Thus, the 5LT system also confirmed the aforementioned advantages in extreme conditions, such as those of the pandemic.

### 4.5. 5-Level Triage in COVID Patients

In the areas dedicated to COVID-19 patients, the waiting times were analogous to the wait times of the general population. During the COVID-19 pandemic, the ED of our hospital had to be deeply reorganized in order to reduce the risk of transmission of the infection as much as possible, and to better direct the flow of patients. During the first phase of the pandemic in 2020, the positive COVID patients were admitted to a specific area that had been created in the Infectious Disease department. The COVID patients who were in need of hospital admission were subsequently referred to specialty inpatient wards which were reserved for positive patients. Further on, when the pressure of the pandemic decreased, the need to create separate flows of patients persisted, and a separated area inside the general ED was rearranged. Simultaneously, the restrictions on visitors and companions (except for special circumstances) were kept in place in order to minimize the potential risk of contagion and retransmission. Screenings with rapid SARS-CoV-2 nasal swabs, which allowed the results to be seen within 6 h, were performed upon presentation to the hospital, and the results were of the utmost importance for identifying and redirecting COVID-positive patients. Despite having process times which were comparable to those of the general population, these patients were subjected to a higher mortality rate.

### 4.6. Strengths and Limitations of the Study

One of the main strengths of this study is the size of the cohort analyzed. Others strong points consist of the fact are that it includes all the causes of access to triage in real life and that it investigates the effects of crowding on the triage system by examining in detail all the variables of crowding, as well as wait times and over- and under-triage. The results of simulations that can be conducted to better manage the flow of patients in the ED are of course important; however, even studies that are carefully tailored will eventually not be representative of the “real life” scenarios when considering data that are obtained over a longer period. This study, conducted on-site, permits the analysis of the events that characterize a “real” clinical cohort, composed of geriatric patients as well as more complex and fragile patients. However, it should be underlined that the study in question has two main major limitations. It is, in fact, a retrospective study, with all the resulting limitations: first of all, the impossibility of selecting patients a posteriori and, therefore, the possibility that the result of the study is modulated by variables that the experimenter cannot control. Furthermore, it is a single-center study, which therefore analyzes the catchment area of our hospital. It will be necessary to carry out multicenter and prospective studies that validate these data.

### 4.7. Future Directions

Our study demonstrates the superiority of the 5-level triage system in our reality. We also underline the importance of multicenter studies representative of the various Italian realities in order to be able to more strongly recommend the use of a 5-level triage system as the standard in the Italian country.

Triage in geriatric people remains a real-life challenge. It is the opinion of the authors that an improvement is possible through the use of artificial intelligence, thus opening up a new field of research.

## 5. Conclusions

The waiting times for geriatric patients, when corrected for triage code, overlap with those of younger patients. With the introduction of the 5-level triage system, geriatric patients, as well as younger patients who required urgent medical examination but did not require high care intensity, have seen reductions in waiting time.

Triage in the geriatric population remains an open challenge for the emergency physician, as these patients are at increased risk of UT and OT.

Increased waiting times have an influence on crowding indices, such as boarding and exit blocks. The worsening of crowding output factors is accompanied by an increased risk of UT. The 5LT already seems to improve the risks of UT and OT triage in crowding conditions.

During the pandemic, and at the same time as a reduction in ED visits, we experienced reduced wait times and increased UT. At the same time, exit blocks and boarding worsened.

## Figures and Tables

**Table 1 jpm-14-00195-t001:** Principal personal and ED presentation features of patients included in the study, by period of observation.

	Period		
	4LT	5LT	*p* ^a^
	N (%)	N (%)	
Sex			
Male	59,432 (51.2)	158,914 (51.7)	
Female	56,628 (48.8)	148,283 (48.3)	0.002
Age			
<75	91,102 (78.5)	234,512 (76.3)	
75+	24,968 (21.5)	72,686 (23.7)	<0.001
Triage priority code			
Code 5	13,443 (11.6)	25,748 (8.4)	
Code 4	78,777 (67.9)	191,981 (62.5)	
Code 3	0 (-)	17,297 (5.6)	
Code 2	22,711 (19.6)	67,688 (22.0)	
Code 1	1129 (0.9)	4484 (1.5)	<0.001
Priority code at discharge			
Code 5	29,240 (25.2)	43,141 (14.0)	
Code 4	73,995 (63.8)	224,039 (72.9)	
Code 3	0 (-)	425 (0.1)	
Code 2	11,952 (10.3)	36,341 (11.8)	
Code 1	873 (0.7)	3252 (1.2)	<0.001
Care intensity			
Low	92,220 (79.5)	235,026 (76.5)	
Medium-to-high	23,840 (20.5)	72,172 (23.5)	<0.001
Outcome			
Discharge	94,701 (81.6)	246,413 (80.2)	
Hospitalization	17,347 (14.9)	51,043 (16.6)	
Transfer	2166 (1.9)	5746 (1.9)	
Left without being seen	1385 (1.2)	2933 (0.9)	
Other	461 (0.4)	1063 (0.4)	<0.001

The 4LT period (T4) spanned from 1 January 2014 to 30 November 2015; the 5LT period (T5) spanned from 1 December 2015 to 31 December 2020; ^a^: χ^2^ test.

**Table 2 jpm-14-00195-t002:** Wait time by period, code at presentation, and age.

		Age < 75 Years				Age ≥ 75 Years			
	Period	N	Median (min)	Interquartile Range (min)	*p* ^a^	N	Median (min)	Interquartile Range (min)	*p* ^a^
Wait time									
Non-urgency									
Code 4	4LT	12,335	51.6	17.9–108.3		1108	57.4	21.4–116.5	
Code 5	5LT	23,379	48.3	17.5–103.8	0.001	2369	50.0	18.2–109.0	0.037
Minor urgency									
Code 3	4LT	64,636	48.4	19.0–102.9		14,141	71.2	30.6–134.1	
Code 4	5LT	156,088	53.1	20.6–115.9	<0.001	35,893	79.4	32.5–151.6	<0.001
Deferrable urgency									
Code 3	5LT	13,403	23.4	12.4–43.9	-	3894	26.9	14.5–48.3	-
High urgency									
Code 2	4LT	13,535	22.5	10.7–47.9		9176	24.7	12.3–51.2	
Code 2	5LT	39,098	31.7	13.4–73.9	<0.001	28,590	33.4	15.1–73.3	<0.001
Emergency									
Code 1	4LT	596	4.6	2.4–9.3		533	5.3	2.6–10.8	
Code 1	5LT	2544	3.6	1.9–7.1	<0.001	1940	5.2	2.6–10.1	0.369

The 4LT period spanned from 1 January 2014 to 30 November 2015; the 5LT period spanned from 1 December 2015 to 31 December 2020. ^a^: Kruskal–Wallis test.

**Table 3 jpm-14-00195-t003:** Distribution of undertriage (UT) and overtriage (OT) percentages during 4LT and 5LT periods, in geriatric and younger patients.

		Period		
Variable		4-Level Triage N (%)	5-Level Triage N (%)	*p* ^a^
Age < 75 years				
	OT			
	No	81,483 (89.4%)	205,378 (87.6%)	
	Yes	9619 (10.6%)	29,134 (12.4%)	<0.001
	UT			
	No	83,837 (92.0%)	215,494 (91.9%)	
	Yes	7265 (8.0%)	19,018 (8.1%)	0.204
Age ≥ 75 years				
	OT			
	No	19,705 (79.0%)	55,553 (76.4%)	
	Yes	5253 (21.0%)	17,133 (23.6%)	<0.001
	UT			
	No	22,898 (91.8%)	66,104 (90.9%)	
	Yes	2060 (8.2%)	6582 (9.1%)	<0.001

The 4LT period (T4) spanned from 1 January 2014 to 30 November 2015; the 5LT period (T5) spanned from 1 December 2015 to 31 December 2020; ^a^: χ^2^ test, UT: under-triage; OT: over-triage.

**Table 4 jpm-14-00195-t004:** Risk of under-triage (UT), by age, triage level period, and presence of access block.

Period	Age	Intensity of Care	Access Block	OR ^a^	95% Confidence Interval	*p*
4-level triage	<75	Low	No	1.00 (ref.)	-	
			Yes	4.37	3.75–5.10	<0.001
		Moderate-to-high	No	1.00 (ref.)	-	
			Yes	1.12	0.35–3.60	0.847
	≥75	Low	No	1.00 (ref.)	-	
			Yes	4.53	3.79–5.43	<0.001
		Moderate-to-high	No	1.00 (ref.)	-	
			Yes	1.18	0.47–2.95	0.724
5-level triage	<75	Low	No	1.00 (ref.)	-	
			Yes	6.92	6.47–7.39	<0.001
		Moderate-to-high	No	1.00 (ref.)	-	
			Yes	0.94	0.63–1.41	0.761
	≥75	Low	No	1.00 (ref.)	-	
			Yes	6.09	5.63–6.59	<0.001
		Moderate-to-high	No	1.00 (ref.)	-	
			Yes	1.82	1.43–2.32	<0.001

^a^: Odds ratios (ORs) estimated by multiple regression analysis adjusted for age and sex.

**Table 5 jpm-14-00195-t005:** Risk of over-triage (OT), by age, triage level period, and presence of access block.

Period	Age	Intensity of Care	Access Block	OR ^a^	95% Confidence Interval	*p*
4-level triage	<75	Moderate-to-high	No	1.00 (ref.)	-	
			Yes	0.23	0.16–0.31	<0.001
	**≥75**	Moderate-to-high	No	1.00 (ref.)	-	
			Yes	0.34	0.27–0.45	<0.001
**5-level triage**	**<75**	Low	No	1.00 (ref.)	-	
			Yes	0.05	0.01–0.36	0.003
		Moderate-to-high	No	1.00 (ref.)	-	
			Yes	0.16	0.15–0.18	<0.001
	**≥75**	Moderate-to-high	No	1.00 (ref.)	-	
			Yes	0.21	0.19–0.23	<0.001

^a^: Odds ratios (OR) estimated by multiple regression analysis, adjusted for age and sex. No presence of OT was observed for low intensity of care during the 4-level triage period or the 5-level triage period for patients ≥75 years of age.

**Table 6 jpm-14-00195-t006:** Trend of crowding indices over the years.

	2014	2015	2016	2017	2018	2019	2020	*p* for Trend
Boarding ^#^	926	1010	1241	1431	1475	2033	4230	
	9.0%	10.1%	11.4%	12.8%	12.8%	18.8%	36.7%	<0.001
Access block ^#^	786	951	1141	1289	1368	2022	3833	
	7.6%	9.5%	10.5%	11.5%	11.9%	18.7%	33.3%	<0.001
Accesses per day	165.8	165.3	170.8	174.4	176.8	175.8	129.8	
Number of accesses	60,512	60,336	62,527	63,662	64,540	64,181	47,500	

^#^ Boarding and access blocks were calculated only for hospitalized patients.

**Table 7 jpm-14-00195-t007:** Selected time variables accounting for crowding, by age and intensity of care.

Wait Time		Age < 75				Age 75+			
		Observations (N)	Median (min)	Interquartile Range (min)	*p* ^a^	Observations (N)	Median (min)	Interquartile Range (min)	*p* ^a^
Low-intensity care									
	No boarding *	18,128	46.5	18.4–104.4		10,603	64.3	27.4–131.1	
	Boarding #	4579	55.6	22.5–129.7	<0.001	2837	75.5	29.5–156.6	<0.001
Medium-to-high care intensity									
	No boarding *	16,716	15.9	6.5–39.3		18,509	21.3	9.5–48.5	
	Boarding #	2100	24.5	10.3–56.0	<0.001	2830	22.0	9.5–53.9	0.041
Low-intensity c are									
	No access block °	265,186	48.4	18.9–106.4		53,915	67.5	27.6–134.4	
	Access block °°	4655	92.7	34.0–182.4	<0.001	3490	113.7	43.8–205.5	<0.001
Medium-to-high care intensity									
	No access block °	46,443	27.2	11.0–27.2		32,584	29.2	12.7–65.3	
	Access block °°	1998	32.0	12.9–85.3	<0.001	2912	29.6	11.4–80.2	0.122

^a^: Kruskal–Wallis test. No boarding. * = Mean number and percentage of patients who did not go to boarding (for example, patients who did not have to wait for a bed). Boarding # = mean number and percentage of patients who underwent boarding. ° No access block: mean number and percentage of patients who did not go to access block. °° Access block: mean number and percentage of patients who experienced boarding.

**Table 8 jpm-14-00195-t008:** Risk of under-triage (UT), by age, triage level period, and presence of boarding.

Period	Age	Intensity of Care	Boarding	OR ^a^	95% Confidence Interval	*p*
4-level triage	<75	Low	No	1.00 (ref.)	-	
			Yes	0.70	0.59–0.83	<0.001
		Moderate-to-high	No	1.00 (ref.)	-	
			Yes	0.99	0.53–1.86	0.987
	≥75	Low	No	1.00 (ref.)	-	
			Yes	0.75	0.60–0.94	0.014
		Moderate-to-high	No	1.00 (ref.)	-	
			Yes	1.38	0.77–2.48	0.279
5-level triage	<75	Low	No	1.00 (ref.)	-	
			Yes	1.01	0.94–1.09	0.753
		Moderate-to-high	No	1.00 (ref.)	-	
			Yes	0.49	0.34–0.69	<0.001
	≥75	Low	No	1.00 (ref.)	-	
			Yes	1.03	0.94–1.12	0.570
		Moderate-to-high	No	1.00 (ref.)	-	
			Yes	1.02	0.81–1.30	0.849

^a^: Odds ratios (ORs) estimated by multiple regression analysis adjusted for age and sex.

**Table 9 jpm-14-00195-t009:** Risk of over-triage (OT), by age, triage level period, and presence of boarding.

Period	Age	Intensity of Care	Boarding	OR ^a^	95% Confidence Interval	*p*
4-level triage	<75	Moderate-to-high	No	1.00 (ref.)	-	
			Yes	1.27	1.01–1.61	0.044
	≥75	Moderate-to-high	No	1.00 (ref.)	-	
			Yes	0.96	0.77–1.19	0.711
5-level triage	<75	Moderate-to-high	No	1.00 (ref.)	-	
			Yes	1.12	0.99–1.26	0.06
	≥75	Low	No	1.00 (ref.)	-	
			Yes	1.03	0.94–1.12	0.570
		Moderate-to-high	No	1.00 (ref.)	-	
			Yes	0.75	0.68–0.84	<0.001

^a^: Odds ratios (ORs) estimated by multiple regression analysis and adjusted for age and sex. No presence of OT was observed for low intensity of care during the 4-levels triage period or in 5-level triage period for patients <75 years of age.

## Data Availability

The raw data supporting the conclusions of this article will be made available by the authors upon reasonable request.

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
