# Peer review of "Geriatric Population Triage: The Risk of Real-Life Over- and Under-Triage in an Overcrowded ED: 4- and 5-Level Triage Systems Compared: The CREONTE (Crowding and R E Organization National TriagE) Study"

_jpm, 2024, doi:10.3390/jpm14020195_

Round 1

Reviewer 1 Report

Comments and Suggestions for Authors

This study is relevant and useful for clinics and personnel from urgency areas. It is well-written and well discussed. So, authors must discuss about other limitations regarding to the design (retrospective) and for other different populations.

Author Response

This study is relevant and useful for clinics and personnel from urgency areas. It is well-written and well discussed. So, authors must discuss about other limitations regarding to the design (retrospective) and for other different populations.

Dear reviewer, we thank you for your appreciation of our work and for the comments that allowed for its improvement.

We have included a paragraph in “strengths and limitations of the study”. For your convenience we report it below.

However, it should be underlined that the study in question has two main major limitations. It is in fact a retrospective study, with all the resulting limitations, first of all the impossibility of selecting patients a posteriori and, therefore, the possibility that the result of the study is modulated by variables that the experimenter cannot control. Furthermore, it is a single-center study, which therefore analyzes the catchment area of our hospital. It will be necessary to carry out multicenter and prospective studies that validate these data

Reviewer 2 Report

Comments and Suggestions for Authors

This is a very interesting observational study that analyzed the impact of triage systems on waiting time and the problems of under or over triage in elderly patients. All parts of the manuscript are generally well written; methodology is well explained. Results are clearly presented. 

My minor comments are: 

 I will suggest modification of the Introducing section: some parts should be rearranged and the objectives of the study should be mentioned in the last part of the Introduction section. 

What are the implications of these findings for everyday practice in the ER? Did authors suggest using a 5-level triage system as a standard in the ED? 

In my opinion, this manuscript can be accepted for publication after suggested minor changes.

Thank you

Author Response

This is a very interesting observational study that analyzed the impact of triage systems on waiting time and the problems of under or over triage in elderly patients. All parts of the manuscript are generally well written; methodology is well explained. Results are clearly presented.

Dear reviewer, we thank you for your appreciation of our work and for the comments that allowed for its improvement..

My minor comments are:

I will suggest modification of the Introducing section: some parts should be rearranged and the objectives of the study should be mentioned in the last part of the Introduction section.

Dear reviewer, we have carried out the suggested rearrangement by placing the objectives at the end of the introduction.

What are the implications of these findings for everyday practice in the ER? Did authors suggest using a 5-level triage system as a standard in the ED?

Dear reviewer, thank you for your comment which allows us to improve our work; in particolary the section Future directions

Below is the paragraph inserted for your convenience.

Our study demonstrates the superiority of the 5 level triage system also in our reality and underlines the importance of multicenter studies representative of the various Italian realities, in order to be able to more strongly recommend the use of a 5 level triage system as standard on the Italian country .